# Sensitivity and specificity of serum soluble interleukin-2 receptor for diagnosing sarcoidosis in a population of patients suspected of sarcoidosis

Laura E. M. Eurelings[1,2]*, Jelle R. Miedema[3], Virgil A. S. H. Dalm[1,2], Paul L. A. van Daele[1,2], P. Martin van Hagen[1,2], Jan A. M. van Laar[1,2], Willem A. Dik[2]

1 Departments of Internal Medicine and Immunology, Section Clinical Immunology, Erasmus University Medical Center, Rotterdam, The Netherlands, 2 Department of Immunology, Laboratory Medical Immunology, Erasmus University Medical Center, Rotterdam, The Netherlands, 3 Department of Pulmonology, Erasmus University Medical Center, Rotterdam, The Netherlands

* l.eurelings@erasmusmc.nl

**Data Availability Statement:** All relevant data are within the paper and its Supporting Information files.

## Abstract

### Background

The soluble interleukin 2 receptor (sIL-2R) has been proposed as a marker of disease activity in patients with sarcoidosis. However, no studies have evaluated whether serum sIL-2R measurement is of use in establishing the diagnosis of sarcoidosis in patients who are suspected of sarcoidosis among other diseases.

### Methods

A cohort study was conducted, consisting of new patients who visited the immunology outpatient clinic and whose serum sIL-2R levels were available before a definitive diagnosis was established between February 2011 and February 2016. All patients underwent standard diagnostic testing for sarcoidosis (e.g. laboratory tests, radiographic and/or nuclear imaging and/or affected site biopsy). This resulted either in the diagnosis of sarcoidosis or the exclusion of sarcoidosis with the diagnosis of another disease. Results of sIL-2R and angiotensin-converting enzyme (ACE) levels, radiographic and nuclear imaging and histology results were collected and definitive diagnoses were recorded. Sensitivity, specificity, the concordance statistic from the receiver operating characteristic curve and Youden's Index were calculated to assess the performance of sIL-2R in the diagnosis of sarcoidosis and were compared to ACE, currently one of the most used diagnostic biomarkers in the diagnosis of sarcoidosis.

### Results

In total 983 patients were screened for inclusion, of which 189 patients met the inclusion criteria. A total of 101 patients were diagnosed with sarcoidosis after diagnostic workup, of whom 79 were biopsy-proven. In 88 patients a diagnosis other than sarcoidosis was made.

**Funding:** The author(s) received no specific funding for this work.

**Competing interests:** The authors have declared that no competing interests exist.

The sensitivity and specificity of serum soluble interleukin 2 receptor levels to detect sarcoidosis were 88% and 85%. The sensitivity and specificity of ACE were 62% and 76%. Receiver operating characteristic curve analysis revealed that sIL-2R receptor is superior to ACE ($p<0.0001$).

## Conclusion

Serum sIL-2R is a sensitive biomarker and superior to ACE in establishing the diagnosis of sarcoidosis and can be used to rule out sarcoidosis in patients suspected of sarcoidosis.

## 1. Introduction

Sarcoidosis is a multisystem disease of unknown origin, characterized by non-caseating granulomas, which can affect almost any organ system. The diagnosis of sarcoidosis is based on clinical and radiographic manifestations and histopathologic detection of non-caseating granulomas in the affected organ, after exclusion of other diseases that may present similarly. [1] Diagnostic tests that can contribute to the diagnosis of sarcoidosis include serum angiotensin-converting enzyme (ACE), conventional chest radiograph, high-resolution chest computed tomography (CT), broncho-alveolar lavage and fluorodeoxyglucose-positron emission tomography (FDG-PET).[2] Undetected sarcoidosis can lead to substantial morbidity. [3, 4]Although ACE is one of the most used diagnostic biomarkers for sarcoidosis, it lacks sensitivity. [5] High sensitivity is needed when a test is used in the initial diagnostic workup, especially when this test is used to rule out the disease.

One of the main immunologic features of sarcoidosis is the influx of CD4+ T-helper (Th)1 cells at sites of active disease. [6] Th 1 cells secrete interleukin (IL) 2, which promotes T-cell proliferation and survival. [7] IL-2 acts via the IL-2 receptor, which consists of the common γ chain (CD132), a β chain (CD122) and an α chain (CD25). CD25 is overexpressed by activated T-cells and regulatory T-cells and can be secreted from the cell membrane in a soluble form; also referred to as soluble IL-2R (sIL-2R.) Hence, sIL-2R is a surrogate marker for T-cell activation. Peripheral blood sIL-2R levels thus reflect the level of T cell activation in an individual and elevated blood sIL-2R levels correlate with disease activity, for instance in patients with rheumatoid arthritis, systemic lupus erythematosus or IgG4-related disease, but also in sarcoidosis patients. [8–12]

There are various theories on the biological action of sIL-2R in the immunopathology of inflammatory diseases. One of the proposed mechanisms of action is that sIL-2R competes with activated T-cells for available IL-2 and thereby inhibits T-cell proliferation. [13, 14] Another proposed function is that sIL-2R binds IL-2, thereby prolonging IL-2 half-life, which enhances its immune-stimulatory effects. [15, 16] On the other hand, it has been proposed that IL-2 can be presented to CD4+ T-cells via the sIL-2R-IL-2 complex, thereby upregulating FOXP3 expression and differentiation into T regulatory cells that subsequently control immune activity. [17]

Up to now, it is unclear whether sIL-2R is produced to combat the sarcoidosis-associated immune activation or whether it has an active role in the pathogenesis of sarcoidosis. It is clear, however, that there is a correlation between sIL-2R and sarcoidosis. Earlier research showed a correlation between sIL-2R and disease activity, the extent of disease and response to treatment. [11, 18–23] In two studies which specifically examined patients with ocular sarcoidosis, serum sIL2-R has been proposed as a diagnostic biomarker. [24, 25] Another study,

conducted in patients with sarcoid skin lesions demonstrated a correlation between serum sIL-2R levels and the number of involved organs. [26] However, no study has evaluated whether the measurement of serum sIL-2R can be useful in diagnosing systemic sarcoidosis in symptomatic patients, whose diagnosis is still uncertain. Thus it remains unknown, whether serum sIL-2R, apart from being a biomarker for disease activity, can be used as a diagnostic tool for sarcoidosis in clinical practice. We hypothesized that serum sIL-2R is a sensitive and specific diagnostic biomarker that can differentiate between sarcoidosis and non-sarcoidosis patients in a cohort of patients suspected of sarcoidosis.

To test this, we conducted a cohort study consisting of new patients in whom sarcoidosis was suspected, but not yet proven or disproven and whose sIL-2R levels were measured before a diagnosis of sarcoidosis or another disease was established. In this cohort, after diagnostic workup and after a definitive diagnosis (sarcoidosis or another disease) was established, we calculated the sensitivity and specificity of sIL-2R in the diagnosis of sarcoidosis and defined the optimal cut-off value and compared this to serum ACE levels.

## 2. Material and methods

### 2.1 Study population

In this study, we included patients visiting the immunology outpatient clinic, in which sarcoidosis, among other diseases, was a part of the differential diagnosis and of whom serum sIL-2R levels were measured before the definitive diagnosis of sarcoidosis or another disease was established. Sarcoidosis was suspected when a patient was referred to the Immunology department with one or more of these symptoms: 1) lung complaints, i.e. shortness of breath, persistent dry cough or chest pains; 2) uveitis or inflammation of the lacrimal glands; 3) skin disorders, i.e. erythema nodosum, lupus pernio, infiltration of scars or tattoo; 4) neurological complaints, i.e. hearing loss, facial nerve palsy or epilepsy; 5) polyarthritis; 6) incidental finding of lymphadenopathy on imaging.

Medical records of patients, whose serum sIL-2R level was assessed at the Erasmus MC between February 2011 and February 2016, were screened for potential inclusion in our study. The following inclusion criteria were applied: availability of medical records, sarcoidosis as part of the differential diagnosis, availability of serum sIL-2R value before a definitive diagnosis was established and a finished diagnostic workup at the immunology outpatient department (Fig 1). Exclusion criteria were: an established diagnosis of sarcoidosis before serum sIL-2R measurement and immunosuppressive medication within the two weeks prior to sIL-2R measurement or use of ACE inhibitors one month prior to ACE measurement. Patients who received immunosuppressive medication were excluded, because immunosuppressive medication can reduce sIL-2R levels. [11, 27] The diagnosis of sarcoidosis relies on consistent clinical and radiological features and histological demonstration of non-caseating granulomas in the affected organ(s), according to the American Thoracic Society/European Respiratory Society/ World Association of Sarcoidosis and other Granulomatous Disorders (ATS/ERS/WASOG) guidelines. [28] In our Sarcoidosis Expertise Centre, sarcoidosis is diagnosed according to these criteria and the diagnosis is verified by at least one pulmonary specialist and at least one clinical immunologist. Patients without histological confirmation of sarcoidosis were, in addition to the evaluation by a pulmonary specialist and a clinical immunologist, evaluated by a, for sIL-2R and ACE blinded, pulmonary specialist who assessed clinical information, radiologic and nuclear imaging. When categorized as sarcoidosis or very probable sarcoidosis the patient was included in the group of patients with a definitive diagnosis of sarcoidosis. Otherwise, the patient was included as a non-sarcoidosis patient.

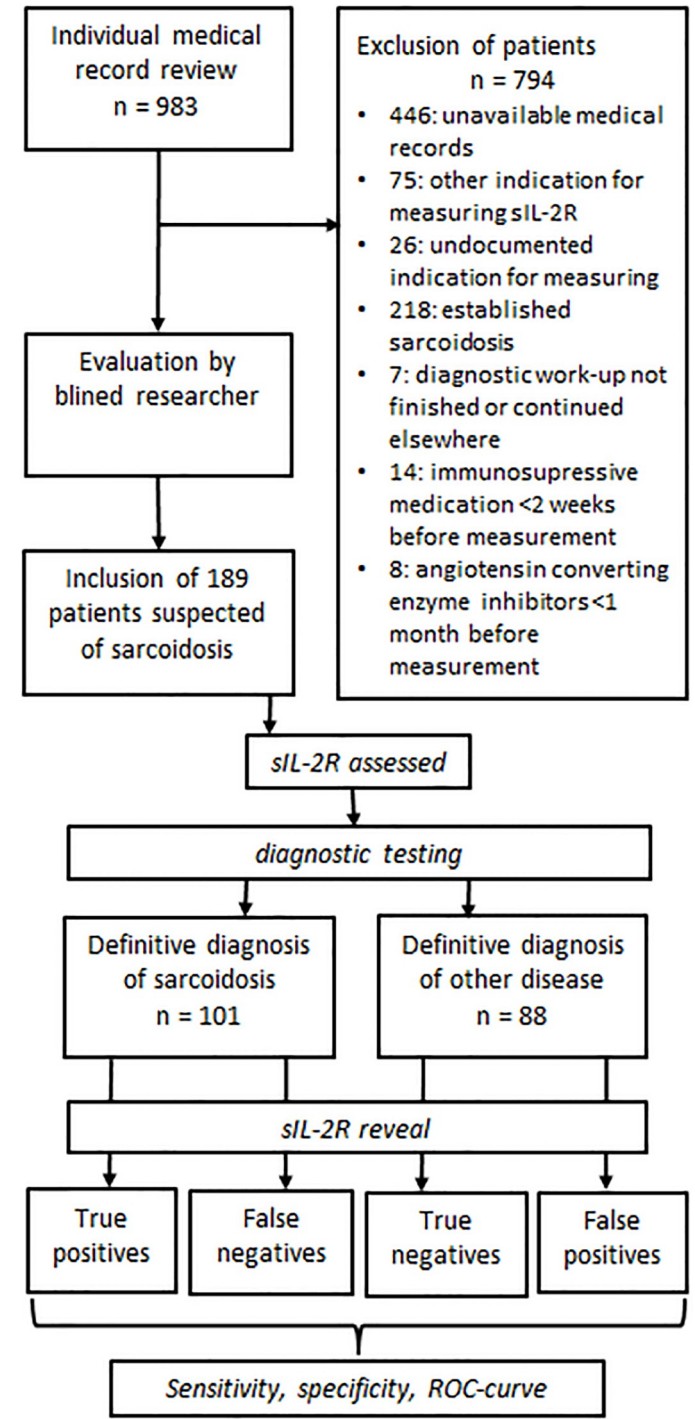

**Fig 1. Flowchart of study design and inclusion of patients.** * sIL-2R: soluble interleukin 2 receptor. ** ROC: receiver operating characteristic curves

The ACE measurement of the date closest to the date of the first sIL-2R measurement was used. If the ACE measurement was performed more than one month prior to or more than one month after sIL-2R measurement, serum ACE was regarded as not available.

Demographic data and anatomical localization of sarcoidosis disease activity were recorded. Four anatomical localizations of sarcoidosis activity were distinguished in this study: sarcoidosis

with predominantly lung, ocular, neurological or skin involvement. Multiple organ involvement could be present within one patient. Additionally, when sarcoidosis was an incidental finding on radiographic imaging, performed for other reasons, with no organ-related symptoms, none of these categories was chosen. Data on chest radiograph, chest CT and FDG-PET scan were collected. Chest radiographs were scored using the Scadding scale. [29]

The serum sIL-2R levels of 101 anonymous blood bank donors were used as healthy controls.

Strengthening the reporting of observational studies in epidemiology guidelines were used to structure the reporting of this observational study.[30] The use of laboratory investigation to gather non-identifiable data in this observational study adheres to the tenets of the Declaration of Helsinki. The medical ethical committee of Erasmus University Medical Center approved the protocol and the associated procedures.

## 2.2 Analysis of serum parameters

Serum sIL-2R measurements were performed (ELISA; Diaclone, Besancon Cedex, France) at the laboratory medical immunology at Erasmus MC. A serum sIL-2R value <2500 pg/mL is considered as not elevated within the Erasmus MC, which is based on values sIL-2R levels measured in 101 healthy blood donors. Serum ACE levels were determined by kinetic assay (Bühlmann Laboratories, Switzerland), with reference range ≤68 U/mL.

## 2.3 Sample size calculation

Hypothetical confidence intervals were calculated using Buderer's formula. Reaching a confidence interval of approximately 5% on either side, with a power of 80%, would require 200 patients Therefore, we strived to include 200 patients.

## 2.4 Statistical analysis

The characteristics of patients with and without the definitive diagnosis of sarcoidosis were summarized using descriptive statistics, such as medians and percentages. Non-parametric tests were used to compare characteristics between the groups. We calculated the specificity, sensitivity and the Youden's index, as well as the concordance statistic (C-statistic), which can be calculated from the receiver operating characteristic (ROC) curve for all patients. Additionally, we calculated the sensitivity, specificity, Youden's index and C-statistic for the subgroup of biopsy-confirmed sarcoidosis patients.

Sensitivity and specificity can be used to calculate the Youden's index. The Youden's index reflects the performance of a dichotomous diagnostic test.

The ROC curve is designed to visualize and detect the optimal performance of a binary test with a varied cut-off point. The optimal cut-off for sIL-2R was calculated by maximizing sensitivity and specificity, using the values generated by a ROC curve. Kruskal-Wallis one-way analysis of variance of the data on the localization of sarcoidosis and the levels of sIL-2R was performed.

The statistical analyses were all done using Excel, IBM SPSS statistics 21.0.0 for Windows (SPSS inc., Chicago, IL, USA), and R, using the package pROC. [31]

## 3. Results

### 3.1 Patient characteristics

Of the 983 screened patients 189 were included and 794 were excluded (Fig 1). The exclusion was due to unavailable medical records (n = 446), sarcoidosis not in the differential diagnosis (thus not being suspected of sarcoidosis at all: n = 75), or undocumented differential diagnosis

(n = 26). Patients were also excluded when the diagnosis was established prior to serum sIL-2R measurement (n = 218) or when the diagnostic workup was not finished or finished elsewhere (n = 7). Furthermore, 22 patients were excluded due to medication use: some patients due to immunosuppressive medication use prior to serum sIL-2R measurement (n = 14), others due to ACE inhibitor use prior to serum ACE measurement (n = 8). Of the 189 included patients, 101 patients were finally diagnosed with sarcoidosis, of which 79 were biopsy-confirmed sarcoidosis patients and another 22 patients without positive biopsy were classified as very probable sarcoidosis by the blinded reviewer. In 88 patients, sarcoidosis was excluded and another disease was diagnosed, including Sjögren's syndrome, tuberculosis, multiple sclerosis or rheumatoid arthritis. A comprehensive list of all the diagnoses is provided in the footnotes of Table 1. ACE levels were available in 120 patients.

Demographic characteristics did not differ between the sarcoidosis and non-sarcoidosis groups (Table 1).

Serum sIL-2R levels are depicted in Fig 2 and presented in Table 1. The median serum sIL-2R level in patients with a definitive diagnosis of sarcoidosis (n = 101) was 6100 pg/mL, 2600 pg/mL in patients with a definitive diagnosis of another disease(n = 88) and 1515 pg/mL in healthy controls (n = 101, healthy blood bank donors). The sIL-2R levels were significantly higher in patients with a definitive diagnosis of sarcoidosis compared to patients with diseases other than sarcoidosis (P < 0.001). In patients with a definitive diagnosis of granulomatous diseases, lung diseases, uveitis, infectious diseases or other diseases, sIL-2R levels were higher than healthy controls (P < 0.001) but lower than sarcoidosis patients (P < 0.001).

**Table 1. Patient characteristics.**

| | Definitive sarcoidosis (n = 101) | Definitive diagnosis other than sarcoidosis[b] (n = 88) |
|---|---|---|
| Sex, female (% female) | 51 (50%) | 52 (59%) |
| Median age at diagnosis, years (IQR[a]) | 43 (35–52) | 44 (31–57) |
| Positive biopsy (n) | 79 | 0 |
| Angiotensin-converting enzyme | | |
| median | 77 | 51 |
| IQR [a] | 44–109 | 31–69 |
| Soluble interleukin 2 receptor | | |
| median | 6100 | 2600 |
| IQR [a] | 4500–9850 | 1925–3300 |

[a] IQR: interquartile range

[b] Including uveitis of unknown origin (n = 19), Nonspecific interstitial pneumonia (n = 3), Systemic lupus erythematosus (n = 3), Asthma (n = 2), Chronic obstructive pulmonary disease (n = 2), Fatigue of unknown origin (n = 2), Fibromyalgia (n = 3), Multiple sclerosis (n = 2), Neuromyelitis optica (n = 2), Ocular vasculitis (n = 2), Rheumatoid arthritis (n = 2), Tuberculosis (n = 2), Sjögren's syndrome (n = 2), Sudden deafness of unknown origin (n = 2), Alopecia (n = 1), Arthritis psoriatica (n = 2), Atypical parkinsonism (n = 1), Auto-immune encephalitis (n = 1), Cervical myelopathy (n = 1), Chronic cough due to reflux (n = 1), Dacryops (n = 1), Epstein Barr viral infection (n = 1), Erythema nodosum (n = 1), Exertional dyspnea (n = 1), Extrinsic allergic alveolitis (n = 1), Folliculitis (n = 1), Guillain-Barre syndrome (n = 1), Herpes simplex related apthosis (n = 1), Human immunodeficiency virus (n = 1), Hyper IgE syndrome (n = 1), Hypophysitis (n = 1), Idiopathic pulmonary fibrosis (n = 1), Increased intracranial pressure (n = 1), Klebsiella pneumoniae infection (n = 1), Langerhans cell histiocytosis (n = 1), Lipoma (n = 1), Neuroendocrine tumor (n = 1), Macular retinopathy (n = 1), Nontuberculous Mycobacterium infection (n = 1), Osteoarthritis (n = 1), Presumed ocular histoplasmosis (n = 1), Rosacea (n = 2), Schwannoma (n = 1), Scleritis (n = 1), Small fiber neuropathy (n = 1), Stills disease (n = 1), Systemic vasculitis (n = 1), Toxoplasmosis (n = 1), Ulcerative colitis (n = 1), Urticaria (n = 1), Vitreomacular traction syndrome (n = 1), Vocal cord paralysis (n = 1)

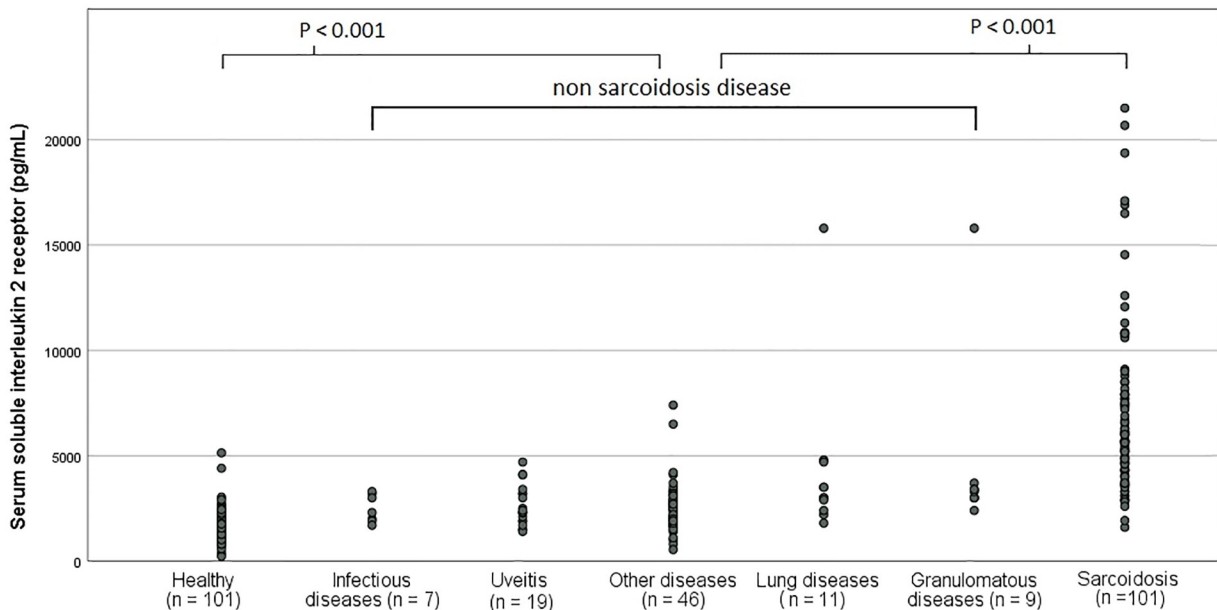

**Fig 2. Dot plot of serum soluble interleukin 2 receptor levels in various diseases and healthy controls.** [*]. Four patients have been assigned towards multiple groups: Two tuberculosis and one extrinsic allergic alveolitis patients were assigned towards lung diseases and granulomatous diseases. One Patient with toxoplasmosis was assigned towards granulomatous diseases as well as towards infectious diseases.

Of all definitive sarcoidosis patients, 65% had a positive chest radiograph, while 99% had a positive CT-scan. Of all sarcoidosis patients who underwent a PET-CT scan, 91% had a positive PET-CT scan. Thirty-five percent of sarcoidosis patients presented with sarcoidosis stage 0 on chest radiograph, while 28% showed stage I, 25% stage II, 10% stage III and only 2% showed stage IV fibrosis on the chest radiograph.

### 3.2 Sensitivity, specificity, Youden's index, and C-statistic

The sensitivity of sIL-2R was superior to ACE (Table 2) in the diagnosis of sarcoidosis. The Youden's index performed better for sIL-2R than for ACE (0.73 *versus* 0.38, respectively). The ROC curve (Fig 3) depicts the performance of both sIL-2R and ACE. The C-statistic, a measure of test performance which can be calculated from this curve, is significantly better for sIL-2R

**Table 2. Sensitivity, specificity, Youden's index and concordance statistic in the diagnosis of sarcoidosis.**

|  | Sensitivity (95% CI[a]) | Specificity (95% CI[a]) | Youden's Index[b] | Concordance statistic[c] (95% CI[a]) |
|---|---|---|---|---|
| sIL-2R, cutoff 2500 pg/mL | 99% (95–100) | 47% (36–57) | 0.45 | 0.91 (0.87–0.96) |
| sIL-2R, cutoff 3550 pg/mL | 88% (82–94) | 85% (78–93) | 0.73 | 0.91 (0.87–0.96) |
| ACE, cutoff 68 U/mL | 62% (51–73) | 76% (64–88) | 0.38 | 0.68 (0.58–0.78) |
| sIL-2R, cutoff 3550 pg/mL, biopsy-confirmed sarcoidosis | 87% (80–95) | 85% (78–93) | 0.73 | 0.93 (0.87–0.98) |
| ACE, cutoff 68 U/mL, biopsy-confirmed sarcoidosis | 60% (48–72) | 76% (64–88) | 0.36 | 0.68 (0.50–0.79) |

[a] CI: confidence interval

[b] Interpretation Youden's index: A higher Youden's index is more favorable.

[c] Interpretation concordance statistic: A higher concordance statistic is more favorable.

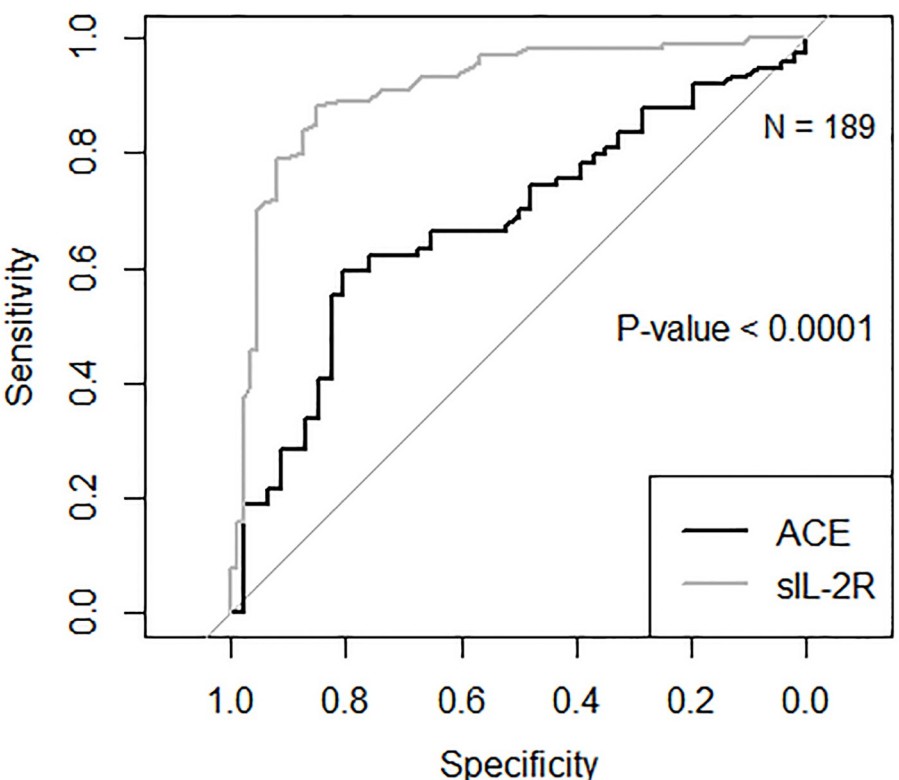

**Fig 3. Receiver operating characteristic curves comparing soluble interleukin 2 receptor levels and angiotensin-converting enzyme in the diagnosis of sarcoidosis (N = 189).** * sIL-2R: soluble interleukin 2 receptor. ** ACE: angiotensin-converting enzyme

(C-statistic = 0,91, with 95% CI 0,87–0,96) *versus* ACE (C-statistic = 0,68 with 95% CI 0.58–0.78) (P < 0.001).

The area under the curve (AUC) is a measure of test performance. A larger area under the curve indicates a better test performance.

When calculating the sensitivity and specificity of serum sIL-2R in a subgroup of patients, consisting of patients with a definitive diagnosis of biopsy-confirmed sarcoidosis (n = 79) and patients with a definitive diagnosis other than sarcoidosis (n = 88), the associated sensitivity and specificity of sIL-2R and ACE similar to the sensitivity and specificity for the whole study group (Table 2). In this subgroup, the C-statistic remained significantly better for sIL-2R *versus* ACE (P = 0.003).

### 3.3 Correlation of sIL-2R with diagnostic parameters

There is a significant, but rather weak correlation between sIL-2R and chest radiograph stage (Spearman's rho 0,24; P = 0.021). Median sIL-2R levels were lower in patients without parenchymal involvement (stages 0 and I) as compared to patients with parenchymal involvement (stages II, III and IV). (Table 3).

Only two patients with a definitive diagnosis of sarcoidosis had no sarcoidosis-associated signs on CT-scan. Therefore, a correlation between CT positive and negative patients was not performed. However, the two patients with a negative CT-scan, displayed markedly lower serum sIL-2R levels (3100 pg/mL and 3700 pg/mL), in comparison to the interquartile range of patients with a positive CT-scan (5700–7900 pg/mL). Although the sIL-2R value of the two

**Table 3. Diagnostic parameters in sarcoidosis patients.**

| | | Number of patients | Median sIL-2R levels (IQR[b]) in pg/mL |
|---|---|---|---|
| Chest radiograph[a] | Stage 0<br>*normal chest radiograph* | 31 | *5500 (3700–7700)* |
| | Stage 1<br>*hilar or mediastinal nodal enlargement* | 27 | *6300 (4854–9100)* |
| | Stage 2<br>*nodal enlargement and parenchymal disease* | 19 | *10800 (5300–19368)* |
| | Stage 3<br>*parenchymal disease without nodal enlargement* | 8 | *5947 (3700–8625)* |
| | Stage 4<br>*pulmonary fibrosis* | 5 | *8487 (5620–15932)* |
| Chest CT-scan | Negative<br>*no sarcoidosis-associated findings* | 2 | *3100; 3700* |
| | Positive<br>*Sarcoidosis-associated findings* | 74 | *6564 (4841–10800)* |
| FDG-PET-scan | Negative<br>*No sarcoidosis-associated findings* | 2 | *3600, 4600* |
| | Positive<br>*Sarcoidosis-associated findings* | 24 | *5700 (4075–7900)* |

[a] Chest radiograph staging: stage 2 is both nodal enlargement and parenchymal disease, while stage 3 is parenchymal disease without nodal enlargement. Therefore, stage 2 is widely regarded as more severe than stage 3.

[b] IQR: interquartile range

FDG-PET negative sarcoidosis patients was lower (3600 pg/mL and 4600 pg/mL) than the median sIL-2R value of the FDG-PET positive sarcoidosis patients (5700 pg m/L), the values were near or within the interquartile range observed of the FDG-PET positive patients (4075–7900).

## 3.4 Determining the optimal cutoff of serum sIL-2R to facilitate the diagnosis of sarcoidosis

The current cut-off used in our hospital to indicate an elevated serum sIL-2R level in an individual is 2500 pg/mL and as such was not chosen to facilitate the diagnosis of a certain disease. This is clearly reflected by the unbalanced specificity sensitivity pattern for sarcoidosis when this sIL-2R cut-off was applied (Table 2). ROC analysis indicated that a serum sIL-2R value of 3550 pg/ml represents the optimal cut-off for diagnosing sarcoidosis with 88% sensitivity and 85% specificity and a Youden's Index of 0.73, which is considered good.

Applying this optimal cut-off to the biopsy-confirmed subgroup resulted in similar sensitivity and specificity and a Youden's index of 0.73. In order to make a fair comparison, we also calculated the optimal cut-off point for ACE, using the ROC curve. The optimal cut-off point for ACE was 68.75, which equals its current cut-off point.

## 3.5 Serum sIL-2R levels in sarcoidosis subgroups

A trend for a higher median sIL-2R level was observed in case of nervous system involvement, however, this was not statistically significant (Table 4).

## 4. Discussion

In this study we demonstrated that serum sIL-2R has high and superior sensitivity and specificity compared to ACE for diagnosing sarcoidosis in a group of patients, presenting with

**Table 4. Soluble interleukin 2 receptor levels.**

| Group | Number of persons | Median sIL-2R level (pg/mL) | IQR[a] |
|---|---|---|---|
| **Healthy controls** | 101 | 1515 | 1150–1880 |
| **Sarcoidosis** | 101 | 6100 | 4500–9850 |
| *Nervous system* | 6 | 16851 | 5027–11020 |
| *Ocular* | 40 | 6150 | 4866–7900 |
| *Skin* | 16 | 6050 | 4975–8750 |
| *Lung* | 48 | 7300 | 4739–11175 |
| **Granulomatous diseases**[b] | 9 | 3300 | 3000–3550 |
| **Lung diseases**[c] | 11 | 3000 | 2400–4700 |
| **Infectious diseases**[d] | 7 | 2300 | 1900–3200 |
| **Uveitis** | 19 | 2300 | 1900–3200 |

[a] IQR: interquartile range

[b] Including Rheumatoid arthritis (n = 2), Tuberculosis (n = 2), Erythema nodosum (n = 1), Extrinsic allergic alveolitis (n = 1), Langerhans cell histiocytosis (n = 1), Systemic vasculitis (n = 1), Toxoplasmosis (n = 1)

[c] Including Nonspecific interstitial pneumonia (n = 3), Asthma (n = 2), Chronic obstructive pulmonary disease (n = 2), Tuberculosis (n = 2), Extrinsic allergic alveolitis (n = 1), Idiopathic pulmonary fibrosis (n = 1)

[d] Including Epstein Barr viral infection (n = 1), Herpes simplex related apthosis (n = 1), Human immunodeficiency virus (n = 1), Klebsiella pneumoniae infection (n = 1), Nontuberculous Mycobacterium infection (n = 1), Presumed ocular histoplasmosis (n = 1), Toxoplasmosis (n = 1)

symptoms, which raised a suspicion of sarcoidosis as well as other diseases. In patients with a definitive diagnosis of sarcoidosis, sIL-2R is significantly elevated compared to patients with a definitive diagnosis of another disease. Thus, in a group of patients who have not yet received a definitive diagnosis, but are suspected of sarcoidosis, serum sIL-2R is able to discriminate between patients with and without sarcoidosis.

Sarcoidosis is a heterogeneous systemic disease and notoriously difficult to diagnose. Evaluation often requires extensive diagnostic testing and invasive tests, such as histological sampling of the affected tissue. [2, 32] The ACE, which is widely used as a diagnostic biomarker for sarcoidosis, has a low sensitivity, ranging from 22–80% that hampers its use in a diagnostic setting. [5, 24, 33, 34] The sensitivity of 60% observed for ACE in the current study is within this range. Our study, using robust and frequently used statistical methods such as the C-statistic and Youden's index, demonstrates that serum sIL-2R is more sensitive and equally specific as ACE in diagnosis in patients suspected of sarcoidosis. [35–39] The demonstrated superiority of serum sIL-2R over ACE as a diagnostic tool for sarcoidosis underlines its use as a valuable clinical instrument in the workup of patients suspected for sarcoidosis and might have consequences for subsequent and potential unnecessary (radiological or invasive) evaluation. [40]

We also observed a significant correlation between serum sIL-2R levels and chest radiograph stage. However, this correlation was rather weak (spearman's' rho 0.24) and might therefore not be clinically relevant. Sarcoidosis stage III chest radiograph represents less severe disease (parenchymal involvement without nodal enlargement) compared to sarcoidosis stage II chest radiograph (parenchymal involvement in addition to nodal enlargement) and was also associated with lower levels of serum sIL-2R than radiographic stage II sarcoidosis. Stage IV disease, which is characterized by pulmonary fibrosis, showed high levels serum of sIL-2R. However, these levels were lower than serum sIL-2R levels found in patients with stage II disease, which is associated with nodal and parenchymal involvement. However, stage IV disease might not necessarily have correlated with disease activity at the time of assessment. Since lung fibrosis is irreversible, stage IV disease can still be visible on chest radiograph, while sarcoidosis

is biochemically and clinically less active. This might explain why the highest values of sIL-2R were not found in stage IV disease, but rather in stage II.

In the tertiary hospital where this study took place, ACE is less expensive than sIL-2R. (€10,00 versus €53,57). However, since sIL-2R is very sensitive, one could speculate that other diagnostic tests, such as CT or FDG-PET might not be necessary for patients with low sIL-2R values. The associated costs (€250,00 and €800,00 respectively) could perhaps be avoided. We measured sIL-2R with ELISA, a relatively rapid (5–6 hours) method that requires standard laboratory equipment and can thus easily be implemented in most clinical laboratories. Alternatively, fully automated techniques are also available. [24] A detailed evaluation of the cost-effectiveness of diagnostic tests used in the diagnostic workup of sarcoidosis is beyond the scope of this study. Yet whether parameters, such as ACE and FDG-PET screening, are rendered redundant by sIL-2R assessment needs evaluation in a cost-effectiveness study

We demonstrate that the probability of having sarcoidosis with a serum sIL-2R level of < 3550 pg/mL is low. This optimal cut-off value of 3550 pg/mL results in a good test performance, both for sensitivity and specificity when used as a diagnostic biomarker for sarcoidosis. [41]

Although underpowered and not designed for subgroup analysis, variations in serum sIL-2R levels between subgroups based on anatomical localization of sarcoidosis disease activity were observed. This was most evident for patients with neurological involvement, who displayed the highest sIL-2R levels. Whether these high serum sIL-2R levels in neurosarcoidosis reflects the total burden of T-cell activation or is an incidental finding, remains unknown.

There is a high clinical variation in sarcoidosis patients. This might be explained by biochemical or genetic differences. It has been shown that certain human leukocyte antigen alleles are associated with variations in the clinical manifestations. For example, the DRB1*0301 allele is associated with Löfgren's syndrome in a Dutch cohort.[42] Whether genetic variances of human leukocyte antigen alleles influence the sIL-2R levels is unknown. Further studies specifically designed for this purpose might further elucidate this observation.

A limitation of our study is the number of patients included. However, the number of included patients (189) did approach our initial sample size calculation of 200 patients. Furthermore, since the effect difference observed in this study between ACE and sIL-2R is larger than the estimated effect difference used in the sample size calculation, it is easier to detect a significant difference. Therefore, our study is adequately powered for our primary outcomes.

A strength of our study is the control group of non-sarcoidosis patients. All patients included in this study were suspected of sarcoidosis at presentation. Therefore, all diseases that could cause sarcoidosis-like symptoms, ranging from tuberculosis to arthritis psoriatica, are included in this study. Therefore, we strongly believe that our study approach most optimally reflects clinical practice

In this study, all sIL-2R levels were measured before the definitive diagnosis was established. Therefore, the outcome, i.e. the diagnosis of sarcoidosis, has not influenced the measurement of sIL-2R and prevents bias. [43] For ACE measurements, only one patient had ACE measured after the diagnosis of sarcoidosis. We feel that this one patient will not have biased the overall result.

There were no missing values for sIL-2R. There were, however, missing values for ACE. When we performed a sensitivity analysis in a subgroup of patients that had no missing values for ACE, sensitivity for sIL-2R was 91% and specificity 85% with a cut-off of >3550 pg/mL. These numbers are almost identical to those of the whole population (88% and 85% respectively). With this, we can safely assume that the missing data are not related to the outcome and therefore the whole study population can be used, including patients who have missing values for ACE.

The results of this study have limitations in their application to clinical practice, due to the population in this study, which does not reflect the general population, but rather the population in a tertiary hospital. The patients in this study were often referred to the tertiary hospital due to difficult to diagnose disease.

This study compared patients with a definitive diagnosis of sarcoidosis and patients with a definitive diagnosis of another disease, not with healthy controls. These patients were diagnosed with various diseases and thus represent a heterogeneous group. Comparing the use of sIL-2R within a diseased population is a comparison that mimics clinical practice more closely than comparing sarcoidosis patients and healthy controls.

## 5. Conclusion

In this cohort study consisting of patients in whom sarcoidosis is suspected but not yet proven or excluded, we demonstrate the superiority of serum sIL-2R above ACE in the screening for sarcoidosis. Therefore, serum sIL-2R may serve as a diagnostic, discriminating tool and as a biomarker for patients that are suspected for sarcoidosis. Due to its high sensitivity, sIL-2R can be a useful tool to rule out sarcoidosis in patients suspected of sarcoidosis.

## Supporting information

**S1 Table. Dataset of sarcoidosis and non-sarcoidosis patients.**
(XLSX)

## Acknowledgments

We would like to thank prof. dr. ir. Eric Boersma for his expertise.

## Author Contributions

**Conceptualization:** Laura E. M. Eurelings, Virgil A. S. H. Dalm, Paul L. A. van Daele, P. Martin van Hagen, Jan A. M. van Laar, Willem A. Dik.

**Data curation:** Laura E. M. Eurelings, Jelle R. Miedema.

**Formal analysis:** Laura E. M. Eurelings.

**Investigation:** Laura E. M. Eurelings, Jelle R. Miedema, Virgil A. S. H. Dalm, Paul L. A. van Daele, Jan A. M. van Laar, Willem A. Dik.

**Methodology:** Laura E. M. Eurelings, Jan A. M. van Laar.

**Resources:** P. Martin van Hagen, Jan A. M. van Laar, Willem A. Dik.

**Supervision:** P. Martin van Hagen, Jan A. M. van Laar.

**Validation:** Jelle R. Miedema, Jan A. M. van Laar, Willem A. Dik.

**Visualization:** Laura E. M. Eurelings.

**Writing – original draft:** Laura E. M. Eurelings, Jan A. M. van Laar, Willem A. Dik.

**Writing – review & editing:** Laura E. M. Eurelings, Jelle R. Miedema, Virgil A. S. H. Dalm, Paul L. A. van Daele, P. Martin van Hagen, Jan A. M. van Laar, Willem A. Dik.

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
