## [Decision Letter · Decision Letter 0]

13 Aug 2019

PONE-D-19-20768

Sensitivity and specificity of the soluble interleukin-2 receptor in biopsy-proven sarcoidosis patients

PLOS ONE

Dear Dr. Eurelings, 

Thank you for submitting your manuscript to PLOS ONE. After careful consideration, we feel that it has merit but does not fully meet PLOS ONE’s publication criteria as it currently stands. Therefore, we invite you to submit a revised version of the manuscript that addresses the points raised during the review process.

The manuscript should be major revised according to the Reviewers' valuable comments.

Respond them appropriately.

We would appreciate receiving your revised manuscript by Sep 27 2019 11:59PM. To enhance the reproducibility of your results, we recommend that if applicable you deposit your laboratory protocols in protocols.io, where a protocol can be assigned its own identifier (DOI) such that it can be cited independently in the future. For instructions see: http://journals.plos.org/plosone/s/submission-guidelines#loc-laboratory-protocols

We look forward to receiving your revised manuscript.

Kind regards,

Masaki Mogi

Academic Editor

PLOS ONE

Journal Requirements:

1. Please include captions for your Supporting Information files at the end of your manuscript, and update any in-text citations to match accordingly. Please see our Supporting Information guidelines for more information: http://journals.plos.org/plosone/s/supporting-information.

Reviewers' comments:

Reviewer's Responses to Questions

**Comments to the Author**

1. Is the manuscript technically sound, and do the data support the conclusions?

Reviewer #1: Yes

Reviewer #2: Partly

2. Has the statistical analysis been performed appropriately and rigorously? 

Reviewer #1: Yes

Reviewer #2: Yes

3. Have the authors made all data underlying the findings in their manuscript fully available?

Reviewer #1: Yes

Reviewer #2: No

4. Is the manuscript presented in an intelligible fashion and written in standard English?

Reviewer #1: Yes

Reviewer #2: Yes

5. Review Comments to the Author

Reviewer #1: Sarcoidosis is a systemic granulomatous disease of unknown etiology. The polarization into T helper type-1 response and its associated cytokines have been reported to play key roles in pathogenesis of this disease. Until now, the diagnosis of sarcoidosis is still difficult and sometimes very problematic, almost bases on a combination of clinical and imaging findings, histological demonstration and exclusion of other capable diseases. So, exploring a marker with high sensitivity and specificity for diagnosis of sarcoidosis is very interesting and meaning with clinical practices. However, serum level of sIL-2R is not a new marker for sarcoidosis with many research studies before in which sensitivity and correlation between this serum level with disease activity and other diagnosis parameters were already confirmed. Because this serum correlates with the level of T-cell activation, identification for the specificity of sIL-2R required a bigger survey from many other diseases and I have a few questions that I would like to discuss below.

1. The new information from your study is identification for the specificity of sIL-2R that required a bigger survey from many non-sarcoidosis diseases. In your manuscript, the number patient is small, they are not representative for the disease populations. If you increase these numbers, your study may be more convincing.

2. In Figure 3, please change the format box-plot to dot-plot, and add the number of each group. It will help to recognize the number of patients involved in this study.

3. In the result 3.3: correlation of sIL-2R with diagnostic parameter, the author showed the significant correlation between sIL-2R and chest radiograph stage, but R=0.27 is not difference. May you check it? And it might be better if you show these data by correlation chart.

Reviewer #2: Review comment PONE-D-19-207

The authors examined the diagnostic utility of sIL-2R and ACE in patients with suspected sarcoidosis patients, and concluded that sIL-2R was useful for such biomarkers. Although the manuscript basically well written, this reviewer feels that there is serious concern that should be addressed.

Major

1. Study population:

To consider sensitivity and specificity of biomarker, study subjects (e.g. sarcoidosis, suspected sarcoidosis, other mimicking disease, healthy subjects) are very important.

A) Suspected of sarcoidosis patients: As mentioned in the introduction, sarcoidosis is systemic disease. Thus, there were many courses in suspected sarcoidosis patients (e.g. lung, heart, liver, skin, nervous). Were all the patients who have lung sarcoidosis in the present study? The authors should clearly define the patients who suspected sarcoidosis. To clarify study subject who suspected sarcoidosis, how about the authors consider to limit the study subjects who have lung sarcoidosis or lung disease.

B) As shown in Figure 2, patients with immunosuppressive medication <2 weeks before sIL-2R measurement or ACE inhibitors < 1 month before ACE were excluded. Are there any backgrounds in this setting period? This reviewer feels that all patients who prescribed immunosuppressive medication should be excluded in the study subjects.

C) In Figure 4, to examine sensitivity and specificity of sIL-2R and ACE, which subjects (e.g. included patients n=180, or include patients n=180 + healthy subjects n=101) did the authors use?

2. Results Figures 1 and 2

Please consider combine Figures 1 and 2, to easy to understand for readers.

Minor

3. Figure 3

Please add the numbers of each group in Figure 3.

6. PLOS authors have the option to publish the peer review history of their article (what does this mean?). If published, this will include your full peer review and any attached files.

Reviewer #1: Yes: Ikuko Ueda-Hayakawa

Reviewer #2: No

---

## [Author Response · Author response to Decision Letter 0]

16 Sep 2019

Dear editor,

Thank you for the opportunity to clarify the message of our manuscript. We have answered the remarks of the reviewers, and we feel that the manuscript, after applying the suggested changes, has improved significantly. In addition to this, we noticed that the objective of our study was not stated clearly enough. Therefore, in addition to our replies and corrections based on the advice of the reviewers, we have shifted the focus of the manuscript slightly to highlight our study population and our objective more clearly. Consequently, we have changed the title to “Sensitivity and specificity of serum soluble interleukin 2 receptor in a population of patients suspected of sarcoidosis”. We hope you will find the manuscript sufficiently improved and suitable for publishing in PlosOne. 

Reviewer #1: Sarcoidosis is a systemic granulomatous disease of unknown etiology. The polarization into T helper type-1 response and its associated cytokines have been reported to play key roles in pathogenesis of this disease. Until now, the diagnosis of sarcoidosis is still difficult and sometimes very problematic, almost bases on a combination of clinical and imaging findings, histological demonstration and exclusion of other capable diseases. So, exploring a marker with high sensitivity and specificity for diagnosis of sarcoidosis is very interesting and meaning with clinical practices. However, serum level of sIL-2R is not a new marker for sarcoidosis with many research studies before in which sensitivity and correlation between this serum level with disease activity and other diagnosis parameters were already confirmed. Because this serum correlates with the level of T-cell activation, identification for the specificity of sIL-2R required a bigger survey from many other diseases and I have a few questions that I would like to discuss below.

1. The new information from your study is identification for the specificity of sIL-2R that required a bigger survey from many non-sarcoidosis diseases. In your manuscript, the number patient is small, they are not representative for the disease populations. If you increase these numbers, your study may be more convincing.

Thank you for your comment. We agree that increasing the number of patients, thus patients who got a definitive diagnosis of sarcoidosis or another disease than sarcoidosis, in our study would add statistical power and value to our study. Therefore, we did go back into the dataset and now added patients in whom a diagnosis (sarcoidosis or another disease) was now established (they were not included in the initially submitted manuscript, as, although serum sIL-2R levels were available at that time, no definitive clinical diagnosis was available then(for example, because histology results were still pending).

In total, we added 9 additional patients, of whom 4 got the definitive diagnosis of sarcoidosis and 5 patients who were diagnosed with another disease. The addition of these extra patients to our study implied recalculations for the whole manuscript, so you will find that all of our numbers have changed slightly, although not significantly. With the addition of these patients, the total number of patients in the revised manuscript now is 189. Prior to the analysis of this study, we have performed a power calculation, with a power of 80% and a confidence interval of approximately 5% on either side, with a modest estimated effect difference of 10%. This yielded a suggested sample size of 200 patients. The number of patients included in this study (189), is close to the suggested sample size. Importantly, since the effect difference appeared larger than the estimated effect difference used in the initial power calculations, it is easier to detect significant differences. Therefore, we feel our study is adequately powered for our primary outcome. 

Adding more patients from another population to add power for our secondary outcomes might induce bias to this study. In this study, we selected all patients from the same pool: patients whose sIL-2R had been assessed because sarcoidosis was part of the differential diagnosis. Therefore, we are limited in the number of patients we can include by our population pool of patients that at that time of serum sIL-2R measurement had a differential diagnosis that included sarcoidosis among other diseases. If we would have selected patients from other departments, such as Rheumatology for example with high suspicion of rheumatoid arthritis yet without any suspicion of sarcoidosis to add to the non-sarcoidosis patient group, we include patients who were never suspected of sarcoidosis. For example, a female patient with a family history of rheumatoid arthritis, with a symmetrical polyarthritis of the hands, morning stiffness and a positive rheumatoid factor or anti-citrullinated protein antibody, would in practice never be tested for sarcoidosis. As the aim of our study was to explore whether serum sIL-2R measurements could be of added value to diagnose sarcoidosis in patients that present at the clinic with features that can be associated with sarcoidosis, as well as other diseases, we feel it would not be statistically sound to include serum the sIL-2R from patients who never have been suspected of sarcoidosis. This would clearly not reflect the daily clinical practice where we want to apply serum sIL-2R measurements in patients who are suspected of having sarcoidosis. Furthermore, it would not answer the research question: can sIL-2R be used in case of suspicion of sarcoidosis? Therefore, we only used this pool of patients suspected of sarcoidosis whose sIL-2R was assessed. 

2. In Figure 3, please change the format box-plot to dot-plot, and add the number of each group. It will help to recognize the number of patients involved in this study.

We agree with the reviewer and have changed the initial box-plot into a dot-pot and added the group sizes to make this figure more insightful.

3. In the result 3.3: correlation of sIL-2R with diagnostic parameter, the author showed the significant correlation between sIL-2R and chest radiograph stage, but R=0.27 is not difference. May you check it? And it might be better if you show these data by correlation chart.

We thank the reviewer for addressing this point. Indeed, a correlation of 0.27, even though it might be statistically significant (P < 0.05) is not a strong correlation and presumably of low clinical value. Therefore, we have changed our wording in that paragraph to ‘There is a statistically significant, weak correlation between sIL-2R and chest radiograph stage’ and we have commented on this in the discussion as well. Since this finding is of low clinical value, we feel it might not be appropriate to use an additional graph for this finding. 

Reviewer #2: Review comment PONE-D-19-207

The authors examined the diagnostic utility of sIL-2R and ACE in patients with suspected sarcoidosis patients, and concluded that sIL-2R was useful for such biomarkers. Although the manuscript basically well written, this reviewer feels that there is serious concern that should be addressed.

Major

1. Study population:

To consider sensitivity and specificity of biomarker, study subjects (e.g. sarcoidosis, suspected sarcoidosis, other mimicking disease, healthy subjects) are very important.

A) Suspected of sarcoidosis patients: As mentioned in the introduction, sarcoidosis is systemic disease. Thus, there were many courses in suspected sarcoidosis patients (e.g. lung, heart, liver, skin, nervous). Were all the patients who have lung sarcoidosis in the present study? The authors should clearly define the patients who suspected sarcoidosis. To clarify study subject who suspected sarcoidosis, how about the authors consider to limit the study subjects who have lung sarcoidosis or lung disease.

Thank you for this insightful comment. As you mentioned, there are many localizations of sarcoidosis and sarcoidosis patients may present with a broad variety of symptoms. This directly translates into a difficulty describing the suspicion of sarcoidosis. In our study, we included a wide range of patients who were diagnosed with sarcoidosis after the diagnosis workup, and not only those presenting with lung complaints. On the one hand, including only those patients presenting only with lung complaints, would have resulted in the inclusion of a more homogenous group. On the other hand, including all patients suspected of sarcoidosis, who eventually were diagnosed with sarcoidosis of any organ, enabled us to include more patients, (and thus gain more statistical power). Importantly, this also better reflects the usefulness of sIL-2R in establishing the diagnosis of sarcoidosis, since not all patients suspected of sarcoidosis will initially present themselves to the clinician with lung complaints. 

Sarcoidosis was suspected when a patient was referred to the Immunology department with one or more of these criteria: 

1) lung complaints i.e. shortness of breath, persistent dry cough or chest pains 

2) uveitis or inflammation of the lacrimal glands 

3) skin disorders i.e. erythema nodosum, lupus pernio, infiltration of scars or tattoos

4) neurological complaints i.e. hearing loss, facial nerve palsy or epilepsy

5) polyarthritis

6) incidental finding of lymphadenopathy on imaging

We have added these criteria to the manuscript to clearly describe our study population. 

B) As shown in Figure 2, patients with immunosuppressive medication <2 weeks before sIL-2R measurement or ACE inhibitors < 1 month before ACE were excluded. Are there any backgrounds in this setting period? This reviewer feels that all patients who prescribed immunosuppressive medication should be excluded in the study subjects.

Thank you for this astute observation. We agree that patients who were prescribed immunosuppressive medication should be excluded from this study. To assess immunosuppressive medication use, the list of medication at first visit to the department was screened. The referral letter was also screened for use of any immunosuppressive medication. We cannot know whether the patient had any immunosuppressive medication prior to the referral letter. For example, one patient might have had corticosteroids prescribed to them by their general practitioner. If this medication has been stopped just before the referral and the referral letter, we might not be aware of this use. Therefore, we defined the immunosuppressive use as two weeks before the first visit and sIL-2R determination, since this is the minimum waiting time between referral and visit to the immunology department. Within this time frame, we are certain that patients have not used any immunosuppressive medication. However, for most patients, the time between the referral letter and the first visit to our clinic is longer, and thus increases the interval of time of which we are sure that they did not use immunosuppressive medication. 

With regards to the use of ACE-inhibitors, we were able to employ a broader strategy. Since this medication is prescribed for long periods of time, and not in short courses as corticosteroids are, we were able to use data not only from the list of medication at first visit and the referral letter, but also from letters from other departments of our Medical Centre, such as the cardiology or ophthalmology department. Therefore we were able to achieve the certainty that patients did not use ACE-inhibitors one month prior to ACE measurement. 

C) In Figure 4, to examine sensitivity and specificity of sIL-2R and ACE, which subjects (e.g. included patients n=180, or include patients n=180 + healthy subjects n=101) did the authors use?

Thank you for your comment. In this figure, we examined the sensitivity and specificity of sIL-2R and ACE in included patients (n = 189) that had a suspicion of sarcoidosis in the differential diagnosis. We choose to use this specific population, since this is the population who, in clinical practice, would undergo diagnostic testing for sarcoidosis. We feel that, if we include healthy controls as well, sIL-2R would over-perform in this setting and would not give realistic values that are applicable for the current medical practice. To clarify, we now have added the numbers patient to this Figure. 

2. Results Figures 1 and 2

Please consider combine Figures 1 and 2, to easy to understand for readers.

We welcome this helpful comment and we have merged Figure 1 and 2 into one figure. 

Minor

3. Figure 3

Please add the numbers of each group in Figure 3.

We have added the group size to make this figure more insightful.

---

## [Decision Letter · Decision Letter 1]

2 Oct 2019

Sensitivity and specificity of serum soluble interleukin-2 receptor for diagnosing sarcoidosis in a population of patients suspected of sarcoidosis

PONE-D-19-20768R1

Dear Dr. Eurelings,

We are pleased to inform you that your manuscript has been judged scientifically suitable for publication and will be formally accepted for publication once it complies with all outstanding technical requirements.

With kind regards,

Masaki Mogi

Academic Editor

PLOS ONE

Additional Editor Comments (optional):

No further comment.

Reviewers' comments:

Reviewer's Responses to Questions

**Comments to the Author**

1. If the authors have adequately addressed your comments raised in a previous round of review and you feel that this manuscript is now acceptable for publication, you may indicate that here to bypass the “Comments to the Author” section, enter your conflict of interest statement in the “Confidential to Editor” section, and submit your "Accept" recommendation.

Reviewer #1: All comments have been addressed

Reviewer #2: All comments have been addressed

2. Is the manuscript technically sound, and do the data support the conclusions?

Reviewer #1: Yes

Reviewer #2: Yes

3. Has the statistical analysis been performed appropriately and rigorously? 

Reviewer #1: Yes

Reviewer #2: N/A

4. Have the authors made all data underlying the findings in their manuscript fully available?

Reviewer #1: Yes

Reviewer #2: No

5. Is the manuscript presented in an intelligible fashion and written in standard English?

Reviewer #1: (No Response)

Reviewer #2: Yes

6. Review Comments to the Author

Reviewer #1: Authors answered the remarks of the reviewers, and they had improved their report significantly. The revised version of the manuscript is acceptable for publication.

Reviewer #2: This manuscript has now much improved.

I have no further comments.

7. PLOS authors have the option to publish the peer review history of their article (what does this mean?). If published, this will include your full peer review and any attached files.

Reviewer #1: No

Reviewer #2: No

---

## [Editor Report · Acceptance letter]

9 Oct 2019

PONE-D-19-20768R1 

Sensitivity and specificity of serum soluble interleukin-2 receptor for diagnosing sarcoidosis in a population of patients suspected of sarcoidosis 

Dear Dr. Eurelings:

I am pleased to inform you that your manuscript has been deemed suitable for publication in PLOS ONE. Congratulations! Your manuscript is now with our production department. 

With kind regards,

on behalf of

Dr. Masaki Mogi 

Academic Editor

PLOS ONE